# CHARACTERSHOT: CONTROLLABLE AND CONSISTENT 4D CHARACTER ANIMATION

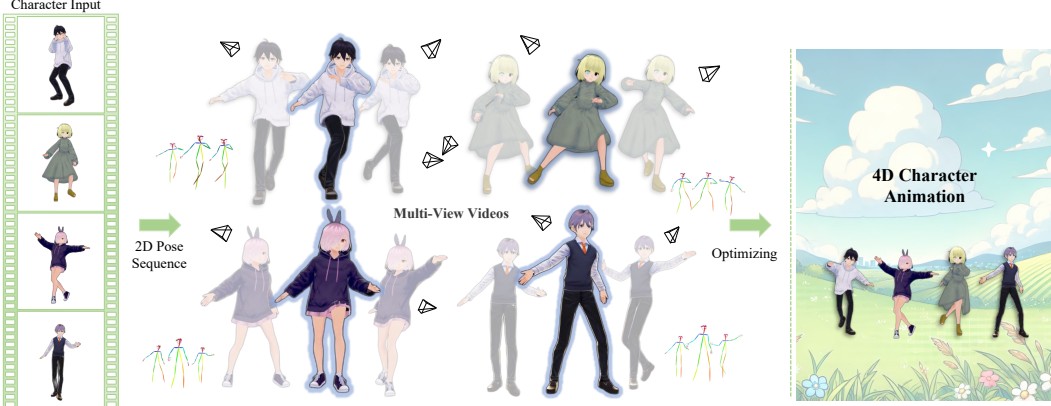

Figure 1: Given any character image and a 2D pose sequence, **CharacterShot** synthesizes dynamic 3D characters with precise motion control and arbitrary viewpoint rendering, achieving both spatial-temporal and spatial-view consistency in 4D space.

## ABSTRACT

In this paper, we propose **CharacterShot**, a controllable and consistent 4D character animation framework that enables any individual designer to create dynamic 3D characters (i.e., 4D character animation) from a single reference character image and a 2D pose sequence. We begin by pretraining a powerful 2D character animation model based on a cutting-edge DiT-based image-to-video model, which allows for any 2D pose sequence as controllable signal. We then lift the animation model from 2D to 3D through introducing dual-attention module together with camera prior to generate multi-view videos with spatial-temporal and spatial-view consistency. Finally, we employ a novel neighbor-constrained 4D gaussian splatting optimization on these multi-view videos, resulting in continuous and stable 4D character representations. Moreover, to improve character-centric performance, we construct a large-scale dataset Character4D, containing 13,115 unique characters with diverse appearances and motions, rendered from multiple viewpoints. Extensive experiments on our newly constructed benchmarks, CharacterBench and HumanBench, demonstrate that our approach outperforms current state-of-the-art methods. Code, models, and datasets will be publicly available.

## 1 INTRODUCTION

When people watch the scientific films such as *The Iron Man*[1] series, they are often amazed by the films' astonishing realism, which leads some to wonder whether such advanced flying suits actually exist in real life. Unfortunately, the answer is *no*, these characters are created by computer-generated imagery (CGI), which includes sophisticated technical chains-from professional 3D modeling and advanced motion capture to complex rigging and retargeting. This CGI pipeline is widely used in film, gaming, and the metaverse, and it requires specialized equipment and significant manual effort

---

[1]https://en.wikipedia.org/wiki/Iron_Man_(2008_film)

to build dynamic 3D characters—a process also known as 4D character animation. In this paper, we introduce CharacterShot, a novel framework that democratizes a low-cost CGI pipeline accessible to individual creators. As shown in Figure 1, CharacterShot supports diverse character designs and custom motion control (2D pose sequence), enabling 4D character animation in minutes and without specialized hardware.

With the remarkable progress in recent generative models (Nichol et al., 2022; Ho et al., 2020), 4D generation (Yin et al., 2023; Zeng et al., 2025; Jiang et al., 2024) has demonstrated the impressive effectiveness in synthesizing 4D content. These methods aim to generate 4D content from a single-view character video. However, they often fall short in practical scenarios—such as those involving hand-drawn or AI-generated characters—where a single-view video including custom motions may not be available. A natural solution is to firstly generate the single-view character video using the 2D character animation methods (Zhang et al., 2025; Ma et al., 2024), which excel at animating a character based on the pose sequence extracted from a target motion video. Such a two-stage framework forms a 4D character animation baseline exhibiting many limitations: 1) Disjoint modeling of pose and view makes it difficult to maintain consistent appearance and motion across views; 2) These 4D methods are trained on general 3D objects from static 3D object datasets such as Objverse (Deitke et al., 2023), suffering from limited diversity in character representations and pose variations—both of which are crucial for generating compelling 4D character animations (Ling et al., 2024; Bahmani et al., 2024; Singer et al., 2023).

To address the above limitations, we propose **CharacterShot**, which is able to generate dynamic 3D characters from a given reference character image and a 2D pose sequence. This flexible and robust 4D character animation requires the model to possess the ability to precisely express the given motion and preserve consistent character appearance across both time and views. To this end, we first enhance the DiT-based image-to-video (I2V) model CogVideoX (Yang et al., 2025c) by integrating pose conditions, enabling user-defined motion control for a given character image. Next, we extend the I2V model to a multi-view setting by introducing a dual-attention module and a camera prior, ensuring both spatio-temporal and cross-view consistency. Finally, we adopt neighboring 3D points as groups with constrained inner-distances within a coarse-to-fine 4D Gaussian Splatting (4DGS) framework to generate a continuous and stable 4D representation from multi-view videos. With these components, CharacterShot produces high-quality and consistent 4D character animation results aligned with the custom motion from 2D pose sequence. Furthermore, to address the scarcity of character-centric 4D animation datasets, we construct a large-scale 4D dataset **Character4D**. Character4D contains 13,115 unique characters with varied appearances, building upon Wang et al. (2024b). Each character undergoes rigging and motion retargeting with diverse 3D motion sequences, followed by multi-view rendering (up to 21 viewpoints), establishing large-scale character-centric 4D dataset specifically designed for 4D character animation.

Moreover, to address the lack of a benchmark for 4D character animation, we establish **Character-Bench**, a benchmark featuring diverse dynamic characters. Extensive qualitative and quantitative comparisons on CharacterBench and a real human benchmark HumanBench demonstrate that CharacterShot outperforms existing state-of-the-art (SOTA) approaches and excels at generating spatial-temporal and spatial-view consistent 4D character animations conditioned on pose inputs. Additionally, ablation studies validate the effectiveness of our framework and highlight its superiority, offering valuable insights to the community. The contributions are summarized as follows:

- To the best of our knowledge, **CharacterShot** is the first DiT-based 4D character animation framework capable of generating dynamic 3D characters from a single reference character image and a 2D pose sequence.

- We propose a novel dual-attention module, which effectively ensuring spatial-temporal and spatial-view consistency in generating multi-view videos.

- A novel neighbor-constrained 4DGS is proposed to enhance the robustness against outliers or noisy 3D points during 4D optimization, resulting in more continuous and stable 4D representations.

- A large-scale character-centric dataset containing 13k characters with high-fidelity appearances rendered with varied motions and viewpoints for 4D character animation.

- Extensive experiments demonstrate that CharacterShot has achieved SOTA performance compared to other methods.

## 2 RELATED WORK

### 2.1 CHARACTER ANIMATION

Recently, with the significant progress in image and video generation made by diffusion models (Ho et al., 2020; Nichol & Dhariwal, 2021; Nichol et al., 2022; Zhao et al., 2025; Li et al., 2024b), numerous character animation methods (Feng et al., 2023; Ma et al., 2024; Chan et al., 2019; Hu, 2024; Zhang et al., 2025; Wang et al., 2025; Luo et al., 2025; Shao et al., 2024; Gan et al., 2025; Tan et al., 2025; Zhu et al., 2024) have exhibited remarkable performance. These works typically generate consistent animation results by using pose skeletons—extracted from off-the-shelf human pose detectors—as motion indicators, and further finetuning U-Net (Ronneberger et al., 2015) or diffusion transformers (DiT) based (Peebles & Xie, 2023) video generation models. In this paper, we build our CharacterShot on the powerful DiT-based image-to-video model CogVideoX (Yang et al., 2025c) to enable higher-quality character animation.

### 2.2 3D GENERATION

Generating 3D content is essential and in high demand across real-world applications. Traditional methods typically rely on 3D supervision to learn 3D representations such as point clouds (Rückert et al., 2022; Kerbl et al., 2023), meshes (Wei et al., 2024; Liu et al., 2024b; Xu et al., 2024), and neural radiance fields (NeRFs) (Hong et al., 2024; Jiang et al., 2023; Tochilkin et al., 2024; Qu et al., 2024). Recent works (Poole et al., 2023; Tang et al., 2024; Shi et al., 2024a; Wang et al., 2024a; Li et al., 2024d; Weng et al., 2023; Pan et al., 2024a; Chen et al., 2024; Sun et al., 2024a; Sargent et al., 2024; Liang et al., 2024; Zhou et al., 2024; Guo et al., 2023; Yi et al., 2023; Yang et al., 2024a) borrow the prior information from 2D image diffusion models, using SDS loss (Poole et al., 2023) to optimize the 3D content from text or image. Other approaches (Liu et al., 2024a; 2023; 2024c; Long et al., 2024; Voleti et al., 2025; Ye et al., 2024; Karnewar et al., 2023; Li et al., 2024a; Shi et al., 2024b; 2023; Wang & Shi, 2023) first generate multi-view images from diffusion models and then perform 3D reconstruction based on these views. In our work, we use the view images generated by a finetuned SV3D (Voleti et al., 2025), as reference view images in the 4D generation stage.

### 2.3 4D GENERATION

Similar to 3D generation, many methods (Yin et al., 2023; Zeng et al., 2025; Jiang et al., 2024; Zhao et al., 2023; Ren et al., 2023; Ling et al., 2024; Bahmani et al., 2024; Singer et al., 2023; Pang et al., 2025) utilize SDS-based optimization to generate 4D content by distilling pre-trained diffusion models in a 4D representation. However, optimizing SDS loss is often computationally intensive and time-consuming. Another line of work (Pan et al., 2024b; Yang et al., 2025b; Zeng et al., 2025; Xie et al., 2025; Sun et al., 2024b; Park et al., 2025; Yang et al., 2025a; Liu et al., 2025c; Hu et al., 2024) finetunes diffusion models to generate multi-view videos and further optimize 4D content. These methods are limited to single-view video-driven generation and often struggle to effectively control the motion specified by the user. Recently, Human4DiT (Shao et al., 2024) introduces SMPL model (Loper et al., 2023) for all views to enable controllable multi-view video generation. However, it does not include 4D optimization stages, and the SMPL pipeline, which involves mesh vertex optimization and SMPL body rendering, is complex and computationally expensive, making it impractical for real-world applications. In contrast, CharacterShot supports simple and convenient 2D pose conditions and is capable of generating spatial-temporal and spatial-view consistent 4D results.

### 2.4 3D/4D CHARACTER GENERATION

Focusing on character-centric 3D/4D generation, many methods learn canonical 3D Gaussian (or mesh) representations with pose-driven deformations, either by optimizing them directly from monocular videos (Li et al., 2024c; Qian et al., 2024; Kocabas et al., 2024; Lei et al., 2024; Hu et al., 2024) or by predicting them in a feed-forward manner from one or a few images (Qiu et al., 2025a;b; Zhuang et al., 2025), in order to construct animatable human avatars by binding them to SMPL models. With the rapid development of large diffusion models, some works (Peng et al., 2024; Huang et al., 2025; Qiu et al., 2025c; Sim & Moon, 2025; Pang et al., 2025; Liu et al., 2025b) leverage multi-view or video diffusion priors to generate pose- and view-rich supervision for human

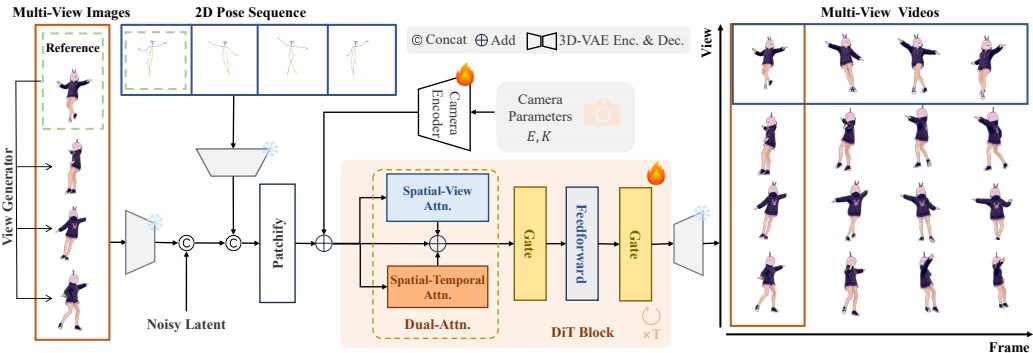

Figure 2: Overview of CharacterShot. Given a reference character image and a 2D pose sequence as custom motion input, our framework generates multi-view videos with spatio-temporal and cross-view consistency. Next CharacterShot apply a neighbor-constrained 4DGS to generate 4D content.

and character avatars, enabling the optimization of 3D/4D Gaussian representations with motions from rigged skeletons or bound SMPL models. However, these methods generate dynamic 3D characters by deforming static canonical avatars along pre-defined motion trajectories within a rigging and rendering pipeline that is complex, tightly coupled, and difficult for individual users. To provide a more user-friendly solution, we propose CharacterShot, which generates high-quality 4D character animation from only a single reference character image and a 2D pose sequence.

## 3 METHOD

Previous studies (Zeng et al., 2025; Xie et al., 2025) optimize 4D representations using single-view character video. However, generating this from a custom character image and corresponding motion control is complex and costly in real-world applications. To address this limitation, we propose CharacterShot, a novel framework that enables pose-controlled 4D character animation from a single reference character image with a 2D driving pose sequence. The overall framework of CharacterShot is illustrated in Figure 2, including pose-controlled 2D character animation (Section 3.2), multi-view videos generation (Section 3.3), and neighbor-constrained 4DGS optimization (Section 3.4). We also introduce the foundational concepts of the DiT model and the detailed illustration of our proposed dataset, Character4D, in Section 3.1 and Section 3.5, respectively.

### 3.1 PRELIMINARIES

In CharacterShot, we utilize a DiT-based image-to-video (I2V) model, CogVideoX (Yang et al., 2025c), as the base model. It consists of a 3D Variational Autoencoder (3D VAE) (Yu et al., 2024), a T5 text encoder (Raffel et al., 2020), and a denoising diffusion transformer (Peebles & Xie, 2023). CogVideoX finetunes a 3D VAE $\mathcal{E}$ to compress both the spatial and temporal information of the input video with the shape $4f \times 8h \times 8w \times 3$ into a latent representation $\mathbf{z_i} = \mathcal{E}(\mathbf{I})$, where $\mathbf{z_i} \in \mathbb{R}^{f \times h \times w \times 16}$. To enable I2V generation, a reference latent $\mathbf{z_r} \in \mathbb{R}^{1 \times h \times w \times 16}$ is concatenated with $\mathbf{z_i}$ along the channel dimension to form the final input $\mathbf{z_0} \in \mathbb{R}^{f \times h \times w \times 32}$, where $\mathbf{z_r}$ will be derived from the latent padding of the reference image. After that, a patchify module is applied to convert the latent $\mathbf{z_0}$ into video tokens $\mathbf{x_0} \in \mathbb{R}^{f \times (\frac{h}{n} \cdot \frac{w}{n}) \times C}$, where $n = 2$ denotes the patch size and $C = 3072$ represents the output channel dimension. And the denoising diffusion transformer $\epsilon_\theta$ is trained by minimizing the Mean Squared Error (MSE) loss $\mathcal{L}$ at each time step $t$, as follows:

$$\mathcal{L} = \mathbb{E}_{\mathbf{x}_t, \epsilon \sim \mathcal{N}(\mathbf{0},\mathbf{I}),\mathbf{c},t}\|\epsilon_\theta(\mathbf{x}_t, \mathbf{c}, t) - \epsilon_t\|^2,$$

where $\mathbf{x}_t$ is the noisy latent at time step $t$, and the gaussian noise $\epsilon_t$ is added to the video latent $\mathbf{z_i}$ before the patchify module. $\mathbf{c}$ is the text condition.

### 3.2 POSE-CONTROLLED CHARACTER ANIMATION

To enable controllable generation on CogVideoX, we treat the pose information as an additional reference and perform 2D character animation pretraining as the base model for the next stage.

Specifically, we utilize 3D VAE to compress pose sequence $P \in \mathbb{R}^{4f \times 8h \times 8w \times 3}$ into pose latent $\mathbf{z_p} \in \mathbb{R}^{f \times h \times w \times 16}$. The pose latent $\mathbf{z_p}$ is then concatenated with the video latent $\mathbf{z_i}$ as a condition, and the reference latent $\mathbf{z_r}$ and the corresponding pose latent $\mathbf{z_{p'}}$ of the reference image are concatenated to provide reference information as follows:

$$\mathbf{z_0} = \text{Concat}\left([\mathbf{z_r},\ \mathbf{z_i}], [\mathbf{z'_p},\ \mathbf{z_p}]\right),$$

where $\mathbf{z_0} \in \mathbb{R}^{(f+1) \times h \times w \times 32}$. During training, we exclude the loss from the reference frame and only update the parameters of diffusion transformer. Moreover, to improve the model's robustness to misaligned pose inputs during animation generation, we select the reference image and its corresponding pose image—originally taken from the first frame of the input video—with those from a randomly selected frame.

### 3.3 MULTI-VIEW VIDEO GENERATION

CharacterShot aims to generate multi-view videos with the shape $V \times (4f + 1) \times 8h \times 8w \times 3$ for 4D optimization, where $V$ represents the number of the target views. We first expand the input latent $\mathbf{z_0}$ from 2D pretraining stage with an additional view dimension:

$$\mathbf{z_0} \in \mathbb{R}^{V \times (f+1) \times h \times w \times 32},$$

where the reference images are taken from different views of the same character at the same time, and the pose latent $\mathbf{z_p}$ from a single view is

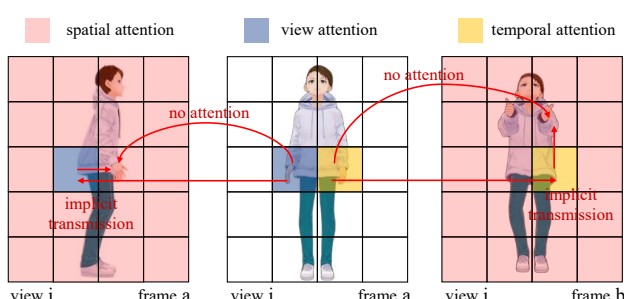

Figure 3: The separated spatial, temporal and view attention mechanisms are difficult to learn the implicit transmission across views and time.

concatenated across all views to enable more adaptive and robust controllable generation. Following SV4D (Xie et al., 2025), the multi-view images are generated by a view generator SV3D (Voleti et al., 2025). We finetune this view generator using our Character4D dataset to improve its performance to characters. Additionally, we encode the camera prior $\pi = (E_v, K_v)_{v=1}^{V}$ into a camera tokens $\mathbf{x_v}$ and add it to the input tokens $\mathbf{x_0} \in \mathbb{R}^{V \times (f+1) \times (\frac{h}{n} \cdot \frac{w}{n}) \times C}$ for each specific view $v$:

$$x_v = \text{rearrange}\left(\mathcal{E}_c(\phi_{\text{plücker}}(E_v, K_v)),\ \left(\frac{h}{n} \cdot \frac{w}{n}\right) \times C\right),$$

where $E_v$ and $K_v$ represent the intrinsic and extrinsic parameters, respectively; $\phi_{\text{plücker}}$ denotes the Plücker embedding (He et al., 2025) with the shape $6 \times 8h \times 8w$; and the camera encoder $\mathcal{E}_c$ encodes the Plücker embedding derived from $E_v$ and $K_v$ into a feature map $C \times \frac{h}{n} \times \frac{w}{n}$.

Previous methods (Xie et al., 2025; Yang et al., 2025b) employ separated spatial, temporal and view attention mechanisms, which are ineffective to learn the implicit transmission of visual information (Yang et al., 2025c), as shown in Figure 3. To address this, we introduce a dual-attention module that includes parallel 3D full attention blocks to model the coherent and consistent visual transmission across spatial-temporal and spatial-view correlations. As shown in Figure 2, we rearrange the tokens $x_0$ with shapes $V \times \left((f+1) \cdot \frac{h}{n} \cdot \frac{w}{n}\right) \times C$ and $(f+1) \times \left(V \cdot \frac{h}{n} \cdot \frac{w}{n}\right) \times C$ as the input to our dual-attention module. We continue training from the 2D pretraining model on our Character4D dataset and initialize the dual-attention module using the weights of its 3D full attention blocks. The synergy of these components enables CharacterShot to generate smooth, spatial-temporal and spatial-view consistent multi-view videos that follow the custom motion defined by the given pose sequences.

### 3.4 NEIGHBOR-CONSTRAINED 4DGS OPTIMIZATION

After obtaining multi-view videos, we apply the neighbor-constrained 4D Gaussian Splatting (4DGS) to optimize the 4D representations. Specifically, we adopt a coarse-to-fine optimization framework followed (Yang et al., 2025a) to model the 4D representations as deformable 3D Gaussians along the temporal axis, with each Gaussian $G$ at time $t$ is represented as:

$$G_t(\mathcal{X}) = G(\mathcal{X}) + F(\gamma(\mathcal{X}), \gamma(t)),$$

where $G(\mathcal{X})$ is the static 3D Gaussians. $F$ is a deformation function and $\gamma(\cdot)$ is a positional encoding function (Tancik et al., 2020).

In the coarse stage, we optimize the static 3D Gaussians $G_{T/2}(\mathcal{X})$ using $\mathcal{L}_1$ loss at $T/2$-th frame, where $T$ denotes the number of frames, to quickly build the initial 4D space first. In the fine stage, we utilize a 4D progressive fitting (Yang et al., 2025a) to gradually refine the deformable Gaussians at time $t$ with the grid-based total variation loss $\mathcal{L}_{\text{TV}}$ (Yang et al., 2025a) and image-space reconstruction losses $\mathcal{L}_1$ and $\mathcal{L}_{\text{LPIPS}}$ from the entire multi-view videos. However, the synthesized multi-view videos might have slight misalignments across views, which often lead to outliers and noisy 3D points during optimization. As shown in Figure 8, previous 4D methods (Yang et al., 2025a; Wu et al., 2024; Yang et al., 2024b; Liu et al., 2025a) results in suddenly disappear hands or visible artifacts. To address this, we introduce a novel neighbor constraint in the fine stage to enforce geometric consistency, which preserves the relative configuration between each 3D point and its neighboring points over time, promoting local deformations, where we select 20 nearest neighbors for each point from the static 3D Gaussians based on the L2 distance. Specifically, we calculate the distances of each 3D point $\mathbf{u}_i$ from the group center at frames $t$ and $t-1$ as:

$$\mathbf{L}_i^t = \mathbf{u}_i^t - \frac{1}{|\mathcal{N}(i)|} \sum_{j \in \mathcal{N}(i)} \mathbf{u}_j^t, \ \mathbf{L}_i^{t-1} = \mathbf{u}_i^{t-1} - \frac{1}{|\mathcal{N}(i)|} \sum_{j \in \mathcal{N}(i)} \mathbf{u}_j^{t-1},$$

where $\mathcal{N}(i)$ represents the neighbor points for $\mathbf{u}_i$. The neighbor loss $\mathcal{L}_{\text{neighbor}}$ is then defined as:

$$\mathcal{L}_{\text{neighbor}} = \sum_{(i,j) \in E} \left\| \mathbf{L}_i^t - \mathbf{L}_i^{t-1} \right\|^2 \cdot w_{ij} \cdot m_{ij}, \ (m_i = \| \mathbf{u}_i^t - \mathbf{u}_i^{t-1} \| > \tau, \quad m_{ij} = m_i \cdot m_j)$$

where $\tau$ is a predefined displacement threshold, $m_{ij}$ is a binary gate that activates only when neighboring points turn into outliers or noisy 3D points, and $w_{ij} = \| \mathbf{u}_i^{t-1} - \mathbf{u}_j^{t-1} \|$ is a spatial edge weight. The full loss function in fine stage can be defined as:

$$\mathcal{L}_{\text{fine}} = \lambda_1 \cdot \mathcal{L}_1 + \lambda_2 \cdot \mathcal{L}_{\text{LPIPS}} + \lambda_3 \cdot \mathcal{L}_{\text{neighbor}} + \lambda_4 \cdot \mathcal{L}_{\text{TV}},$$

where the coefficients $\lambda_1$, $\lambda_2$, $\lambda_3$, and $\lambda_4$ are the corresponding weighting factors.

## 3.5 CHARACTER4D

Current 4D character datasets (Yu et al., 2021b; Cheng et al., 2023) only include a very small variety of character types and motion types. To enable a more generalized 4D character animation, we construct a large-scale 4D character dataset by filtering high-quality characters from VRoidHub[2] (VRoid, 2022)—a platform for sharing and showcasing 3D character models—and collect a total of 13,115 characters in OBJ file format. First, we load the characters into Blender[3], a widely used 3D modeling software, with an initial configuration: A-pose[4] and a centered camera positioned at a fixed height, with the radius and field of view (FoV) set to 2.5 and $40°$, respectively. After that, we bind

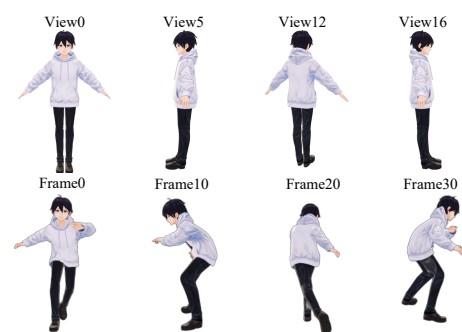

Figure 4: A character sample from our Character4D dataset shown across four views and frames.

40 diverse motions (e.g., dancing, singing, and jumping) in skeletons collected from Mixamo[5] (mix) to these characters, following the data curation pipeline used in previous methods (Chen et al., 2023; Peng et al., 2024; Wang et al., 2024b). Specifically, we assign one randomly selected motion to each character (Wang et al., 2024b) using the automatic retargeting software Rokoko (rok). Binding motion using skeletons helps the clothing swing naturally with the movements, allowing the model to

---

[2] All the 3D avatars we used in our dataset clearly show the permission of usage in their individual websites.

[3] https://www.blender.org/

[4] A standard initial posture in which the character stands upright with arms slightly angled downward and outward, forming an "A" shape.

[5] An online platform by Adobe that provides automatic rigging and a large library of motions.

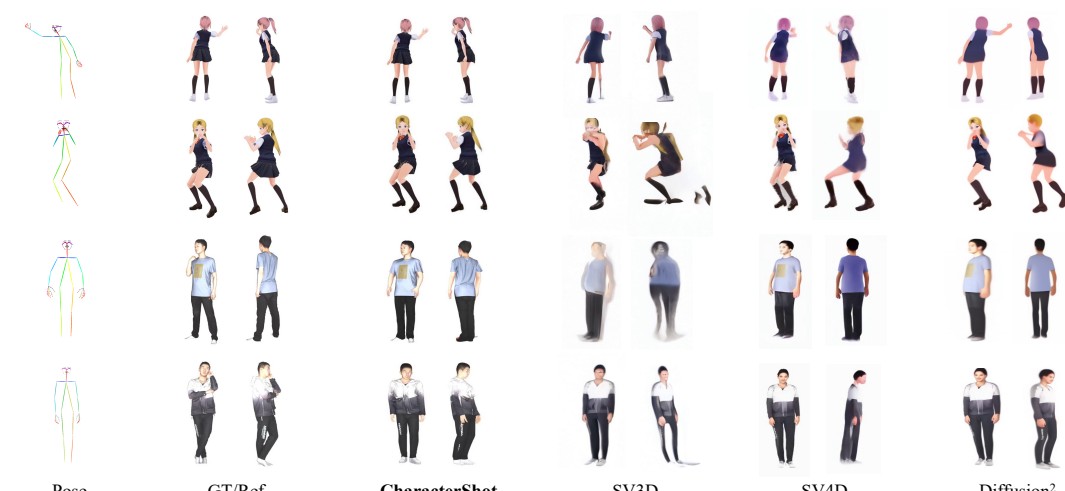

| | Pose | GT/Ref | **CharacterShot** | SV3D | SV4D | Diffusion[2] |

Figure 5: Visual comparison of multi-view videos synthesis. Rows 1–2 show characters from CharacterBench, with the ground-truth images in column 2. Rows 3–4 show humans from HumanBench, and the corresponding static reference images are also shown in column 2. CharacterShot generates high-quality character videos with both spatial-temporal and multi-view consistency, faithfully preserving the reference character image and driving pose.

Table 1: Quantitative comparison of multi-view videos synthesis on CharacterBench. The best result is marked in **bold**.

| Methods | SSIM ↑ | LPIPS ↓ | CLIP-S ↑ | FVD-F ↓ | FVD-V ↓ | FVD-D ↓ | FV4D ↓ |
|---|---|---|---|---|---|---|---|
| SV3D | 0.873 | 0.241 | 0.864 | 1639.020 | 1471.051 | 1378.806 | 2078.984 |
| Diffusion[2] | 0.889 | 0.135 | 0.878 | 1198.645 | 1044.424 | 994.202 | 1392.323 |
| SV4D | 0.891 | 0.138 | 0.856 | 1280.620 | 1537.853 | 1467.422 | 1477.972 |
| CharacterShot | **0.967** | **0.021** | **0.957** | **469.677** | **489.963** | **388.797** | **490.457** |

learn the principles of physical reality. Next, we generate 21 camera viewpoints along a horizontal static trajectory, following the setup used in SV3D (Voleti et al., 2025). Finally, we render frames of all characters from 21 viewpoints in the A-pose for view generator finetuning, and with various motions for diffusion transformer finetuning to generate spatial-temporal and spatial-view consistent multi-view videos from any reference character image and custom motion in pose sequence. We provide the visual examples of our Character4D dataset in Figure 4. The top row shows the character in the A-pose, while the bottom row depicts the character performing a specific motion.

## 4 EXPERIMENTS

### 4.1 IMPLEMENTATION DETAILS

**Benchmarks.** As with the dataset challenges faced by existing 4D generation methods, there is currently no character benchmark for evaluating 4D character animation. To address this, we introduce a new benchmark CharacterBench built from the test sets of Character4D (which comprise 23 characters disjoint from the training data), together with 10 characters that are curated from Mixamo. Characters in the A-pose are used to assess the view generator's performance, while characters with motion are used to evaluate the effectiveness of 4D character animation. Moreover, we construct a HumanBench consisting of 48 real humans collected from the open-source People Snapshot (Alldieck et al., 2018) and THuman 2.1 (Yu et al., 2021a) datasets. Note that each person in HumanBench is provided only as a static 3D model and cannot be used to render multi-view videos as ground truth. Therefore, we animate these humans using the motions from CharacterBench and conduct a user study as a proxy for quantitative evaluation. To evaluate the generalization of CharacterShot, we also select characters that are out-of-Character4D, gathered additional examples from the Internet, and generated a suite of virtual characters using Flux (Labs, 2024), spanning 2D anime

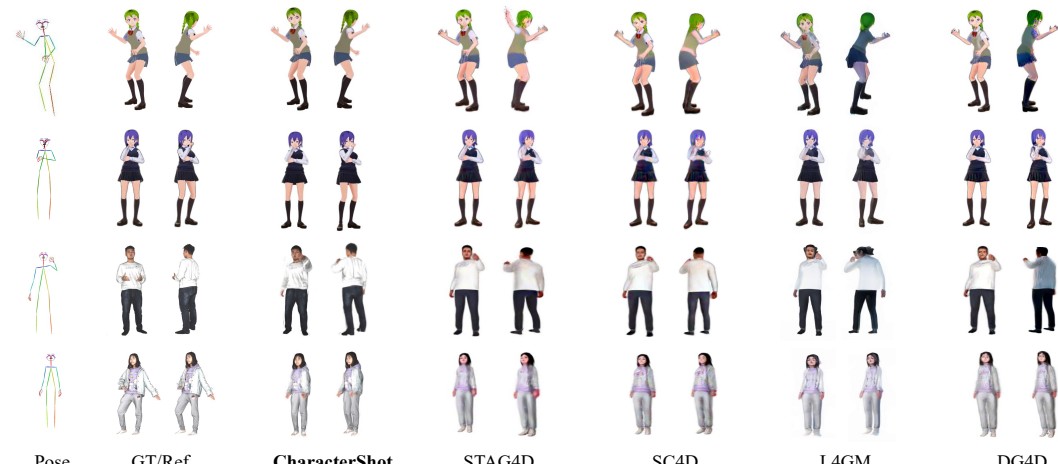

| Pose | GT/Ref | **CharacterShot** | STAG4D | SC4D | L4GM | DG4D |

Figure 6: Visual comparison of 4D generation. Rows 1–2 show characters from CharacterBench, with the ground-truth images in column 2. Rows 3–4 show humans from HumanBench, and the corresponding static reference images are also shown in column 2. CharacterShot outperforms other methods in terms of texture and detail.

Table 2: Quantitative comparison of 4D generation on CharacterBench. The best result is marked in **bold**.

| Methods | SSIM ↑ | LPIPS ↓ | CLIP-S ↑ | FVD-F ↓ | FVD-V ↓ | FVD-D ↓ | FV4D ↓ |
|---------|--------|---------|----------|---------|---------|---------|--------|
| STAG4D | 0.915 | 0.082 | 0.904 | 966.979 | 876.033 | 817.523 | 970.241 |
| SC4D | 0.907 | 0.089 | 0.907 | 961.941 | 849.578 | 813.812 | 995.497 |
| L4GM | 0.907 | 0.091 | 0.892 | 1056.498 | 889.114 | 846.307 | 1042.443 |
| DG4D | 0.888 | 0.116 | 0.897 | 1006.051 | 1200.049 | 1171.713 | 1059.921 |
| CharacterShot | **0.971** | **0.025** | **0.959** | **368.235** | **289.279** | **271.886** | **406.624** |

characters, real-world humans, and other distinct 3D models with diverse motions. Results of these out-of-Character4D test samples are presented in Section B.4 and Figure 14, Appendix.

**Evaluation Metrics.** To verify the effectiveness of our Character4D in improving the performance of fine-tuned view-generator, we follow the protocols of (Voleti et al., 2025; Liu et al., 2023; Xu et al., 2024; Yang et al., 2024a) and use PSNR (Lim et al., 2017), SSIM (Wang et al., 2004), and LPIPS (Zhang et al., 2018) to evaluate the quality and similarity between the generated view images and the ground-truth images from low-level. Also, CLIP-score (CLIP-S) and FID (Heusel et al., 2017) are employed to evaluate

Table 3: User study on HumanBench comparing CharacterShot with baselines. Methods at the top are multi-view video generation methods, and those at the bottom are 4D generation methods.

| Methods | Appearance ↑ | Pose ↑ | Time ↑ | View ↑ |
|---------|--------------|--------|--------|--------|
| SV3D | 12.15 | 9.75 | 10.90 | 8.16 |
| Diffusion[2] | 25.42 | 27.88 | 22.80 | 29.94 |
| SV4D | 19.92 | 23.22 | 25.43 | 19.07 |
| CharacterShot | **42.51** | **39.15** | **40.87** | **42.83** |
| SC4D | 8.64 | 9.90 | 12.23 | 6.56 |
| STAG4D | 17.82 | 19.38 | 18.43 | 14.80 |
| L4GM | 23.98 | 16.10 | 19.67 | 15.61 |
| DG4D | 16.21 | 17.94 | 14.48 | 20.16 |
| CharacterShot | **33.35** | **36.68** | **35.19** | **42.87** |

high-level semantic consistency. For multi-view video generation and 4D optimization, we follow SV4D (Xie et al., 2025) and apply FV4D, FVD-F, FVD-V, and FVD-D to evaluate consistency across frames and views. Visual quality is further evaluated using CLIP-S, LPIPS, and SSIM metrics. We also conduct a user study with 30 participants to assess the consistency of appearance, pose, time, and view in the HumanBench evaluations on 20 samples. Specifically, we ask the volunteers to rank all methods for each sample and assign weighted scores based on the resulting rankings.

## 4.2 COMPARISON WITH SOTA METHODS

**Multi-View Videos Synthesis.** As mentioned in Section 1, previous 4D generation models require single-view videos and are unable to be conditioned on custom motion such as pose sequences.

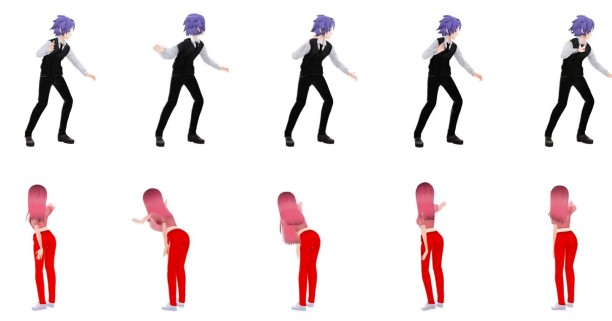

To enable a fair comparison, we adopt a two-stage generation for these methods by finetuning the SOTA 2D character animation model MimicMotion (Zhang et al., 2025) on our collected high-quality 2D pose-driving dataset to generate single-view videos based on each specified character and corresponding pose input. We then compare the proposed CharacterShot with SOTA single-view video-driven 4D generation methods, including SV3D (Voleti et al., 2025), SV4D (Xie et al., 2025) and Diffusion$^2$ (Yang et al., 2025b). We first present the qualita-

Ground Truth (a)Baseline (b)+Camera Prior (c)+Dual Attn (d)w/ View Attn

Figure 7: Visualization from the baseline to variants incorporating different model components.

tive comparison in Figure 5. It is evident that Diffusion$^2$ and SV4D generate results with inconsistent poses across different views (see rows 1, 2 and 3). Notably, all these baselines generate blurred or incorrect details in both the facial and body regions. Thanks to our proposed dual-attention module—which explicitly models both spatial-temporal and spatial-view consistency with camera priors—CharacterShot generates more coherent results with consistent, high-quality details across poses, frames and views in both characters and real-world humans. Quantitative results in Table 1 further verify the effectiveness of the proposed CharacterShot. Specifically, CharacterShot achieves the highest SSIM, LPIPS, and CLIP-S scores, demonstrating strong identity preservation and indicating superior image quality. Additionally, the proposed dual-attention module contributes to the best performance on FVD-F, FVD-V, FVD-D, and FV4D, highlighting its effectiveness in providing high-quality videos and maintaining spatial-temporal and spatial-view consistency. Also, the user study in Table 3 demonstrates that CharacterShot generalizes well to these human inputs, outperforming all baselines on HumanBench in terms of appearance, pose, time, and view consistency. More results of **unseen and out-of-Character4D test samples** from Flux and Internet are presented in Section B.4 and Figure 14, Appendix. Multi-view videos are shown in Supplementary Material.

**4D Generation.** We also present the comparison between SOTA 4D generation methods, including STAG4D (Zeng et al., 2025), SC4D (Wu et al., 2025), L4GM (Ren et al., 2024), and DG4D (Ren et al., 2023)—with our CharacterShot by rendering images in specific 9 views after 4D optimization, while the optimization stage for SV4D and Diffusion$^2$ is not open source. As the qualitative comparison shown in Figure 6, we notice that the results of STAG4D and SC4D exhibit inconsistent shapes and textures (e.g., the left hand and clothing in row 1, 3 and 4), while DG4D suffers from

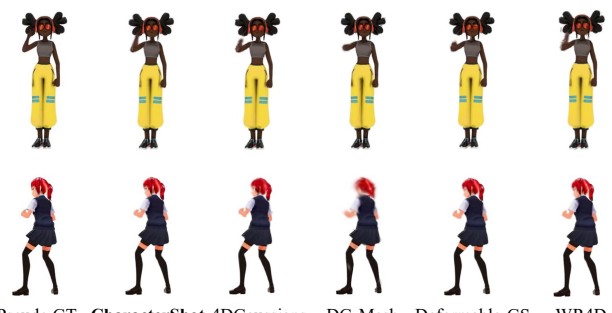

Pseudo GT **CharacterShot** 4DGaussians DG-Mesh Deformable-GS WR4D

Figure 8: Visual comparison of 4D optimization. "Pseudo GT" refers to the multi-view videos produced in the preceding stage.

flickering artifacts. L4GM generates clearer details compared to these three SDS loss-based methods, but it has some black artifacts. In contrast, our CharacterShot generates consistent and continuous high-quality 4D contents by applying dual-attention module and neighbor-constrained 4DGS. The quantitative experiments in Table 2 and user study in Table 3 further demonstrate that our method consistently outperforms the baselines across all metrics. Videos of 4D contents are shown in Supplementary Material.

## 4.3 ABLATION STUDIES

**Contribution Decomposition of Model Components.** We finetune our pretrained 2D character animation model on Character4D and generate videos for each view separately as a single-view baseline, then investigate the impact of our proposed components in the following analysis. As shown in Figure 7(a), the baseline struggles to transform the pose sequence accurately across different viewpoints, leading to noticeable distortions. By incorporating the camera prior, the single-view model achieves more accurate viewpoint-aware pose alignment, resulting in more reasonable position (see Figure 7(b)). The visual results in Figure 7(c) effectively follow the reference's appearance and pose, demonstrating the necessity of simultaneously generating multi-view videos and the effectiveness of our dual-attention module.

Table 4: Quantitative results on model components. "w/ View-Attention" indicates that we use separate view attention as a replacement for our spatial-view attention in dual-attention module.

| Methods | SSIM ↑ | LPIPS ↓ | FVD-F ↓ | FV4D ↓ |
|---|---|---|---|---|
| Baseline | 0.956 | 0.032 | 614.010 | 639.733 |
| + Camera Prior | 0.961 | 0.029 | 545.662 | 570.046 |
| + Dual-Attention | **0.967** | **0.021** | **469.677** | **490.457** |
| w/ View-Attention | 0.964 | 0.025 | 491.865 | 520.737 |

Moreover, to further verify the importance of modeling implicit spatial-view information—rather than treating view information separately—we compare the spatial-view attention with a separate view-attention mechanism. As shown in Figure 7 (c)(d), our dual-attention module with spatial-view attention achieves better performance, demonstrating its superiority in enhancing spatial-view consistency. The experiments in Table 4 further support the visual observations and demonstrate the effectiveness of each component in our framework.

**4DGS Optimization.** To verify neighbor-constrained 4DGS' effectiveness, we compare it with SOTA 4DGS methods 4DGaussians (Wu et al., 2024), WR4D (Yang et al., 2025a), Deformable-GS (Yang et al., 2024b) and DG-Mesh (Liu et al., 2025a). For a fair comparison, we optimize the 4D representations of these methods using our generated multi-view videos (as pseudo ground truth).

Table 5: Quantitative comparison of 4D optimization on CharacterBench. Ground truths are the multi-view videos produced in the preceding stage.

| Methods | SSIM ↑ | LPIPS ↓ | FVD-F ↓ | FV4D ↓ |
|---|---|---|---|---|
| 4DGaussians | 0.984 | 0.017 | 89.726 | 66.962 |
| WR4D | 0.985 | **0.015** | 80.651 | 59.509 |
| Deformable-GS | 0.979 | 0.025 | 194.451 | 198.861 |
| DG-Mesh | 0.980 | 0.023 | 154.596 | 168.652 |
| CharacterShot | **0.987** | **0.015** | **73.284** | **55.472** |

As shown in Figure 8, sudden hand disappearance can be observed in the first row for 4DGaussians, Deformable-GS, and DG-Mesh. In addition, outlier and noisy 3D points also result in blurring and artifacts on the face and body for these methods. In contrast, CharacterShot produces continuous and stable 4D content by applying the neighbor constraint. The quantitative results in Table 5 further validate the effectiveness of our proposed neighbor-constrained 4DGS method.

## 5 CONCLUSION

In this work, we propose CharacterShot, a controllable and consistent 4D character animation framework that generates dynamic 3D characters from just a single reference image and a 2D pose sequence. By leveraging the powerful DiT-based I2V model CogVideoX, CharacterShot first constructs a pose-controlled 2D character animation. Subsequently, CharacterShot introduces a dual-attention module to model implicit visual transmission across views and time, along with a camera prior to help transform pose positions. Finally, a neighbor-constrained 4DGS is employed to generate continuous and stable 4D representations. To further enhance character performance, we construct a large-scale dataset, Character4D, containing 13,115 high-quality characters with corresponding diverse motions. Extensive experiments on CharacterBench and HumanBench demonstrate the advantages of our method in capturing character and human details and achieving both spatial-temporal and spatial-view consistency. We hope that CharacterShot, along with its models and datasets, will contribute valuable and affordable resources to any individual creator and researcher to advance 4D character animation.

## ETHICS STATEMENT

In developing CharacterShot, a controllable and consistent 4D character animation framework that enables any individual designer to create dynamic 3D characters (i.e., 4D character animation) from a single reference character image and a 2D pose sequence, we are dedicated to upholding ethical standards and promoting responsible AI use. During building Character4D dataset, we strictly follow the data curation pipeline of HumanVid (Wang et al., 2024b), and the assets used in Character4D explicitly state permission for use on their respective websites. Our code, model and dataset will be publicly released to encourage responsible use in areas like entertainment and education, while discouraging unethical practices, including misinformation and harassment. We also advocate for continued research on safeguards and detection mechanisms to prevent misuse and ensure adherence to ethical guidelines and legal frameworks.

## REPRODUCIBILITY STATEMENT

To facilitate replication, we provide additional technical details in Appendix A, including the base models and training parameters used at each stage. The data curation pipeline of the proposed Character4D is described in Section 3.5. All code, models, and datasets will be released publicly.

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

APPENDIX

## A  IMPLEMENTATION DETAILS

In the pose-controlled 2D character animation pretraining stage, we initialize our DiT model weights using the pretrained image-to-video model CogVideoX-I2V-5B (Yang et al., 2025c). The pretraining dataset comprises 21,000 dancing videos collected from the Internet, which are processed into 336,000 video clips, each containing 25 frames at a resolution of $480 \times 720$. Next, we apply the widely used pose detector DWpose (Yang et al., 2023) to extract pose images. We follow the full training script from CogVideoX, using a learning rate of 2e-5, and train this stage for 11,000 steps on eight A800 GPUs. In the multi-view video generation stage, we continue finetuning the model on Character4D with dual-attention module and a camera encoder, starting from the checkpoint obtained in the first stage. During training, we set $V = 5$ and randomly sample views from the view pool. This stage is trained for 1,500 steps on 16 A800 GPUs with a learning rate of 5e-5. We also finetune the view generator from SV3D using the Character4D dataset with A-pose, training for 20,000 iterations on eight A800 GPUs at a resolution of 768×768, with each sample consisting of 21 frames. Please note that the view-generator is a plugin component that allows us to seamlessly replace SV3D with any more powerful view-generator at no additional cost.

We finetune MimicMotion on our 2D pretrained dataset to improve its performance on characters, and we only update the parameters of temporal layers and pose guider at (lr=1e-4, batch size=8, gpus=8, resolution=1024, num frames=15, training steps=30000). For neighbor-constrained 4DGS, both the coarse stage and each progressive step (Yang et al., 2025a) in the fine stage are trained for 3000 iterations. In the coarse stage, we select the video frame at time step $T/2$ to optimize a static Gaussian representation. In the fine stage, we utilize the full multi-view video sequence for progressive optimization. For the $\mathcal{L}_{\text{neighbor}}$, we define the local neighborhood of a point as its 20 nearest neighbors in the static 3D Gaussians. For loss weighting, we set $\lambda_2 = 0.01$, while all other coefficients $\lambda_{1,3,4} = 1$. The learning rate is 1.6e-4.

**Metrics.** Folloing, SV4D (Xie et al., 2025), for FV4D, we compute the Fréchet Video Distance (FVD) (Unterthiner et al., 2019) over all images, which are traversed in a bidirectional raster pattern. In addition, we employ three specialized FVD variants to evaluate video coherence at a more granular level: FVD-F, which computes FVD across frames within each view; FVD-V, which computes FVD across views for each frame; and FVD-D, which computes FVD across the diagonal elements of the view–frame matrix. Specifically, we generate 21 views for evaluating the view generator. FV4D, FVD-F, FVD-V, and FVD-D are computed from a $9 \times 9$ multi-view video matrix, which consists of nine viewpoints and nine frames.

## B  EXPERIMENTS

### B.1  DIFFERENT SETTINGS ON 4D OPTIMIZATION

In this subsection, we conduct an ablation study on our neighbor loss and its corresponding binary gate in neighbor-constrained 4DGS. As shown in Table 6, without the full neighbor loss leads to a notable drop in performance metrics, with FV4D and FVD-F suffering the most, showing over 10% degradation. Moreover, only removing the binary gate in the neighbor loss also results in performance degradation, whereas using the full setting achieves the best results across all metrics.

Table 6: Ablation study for our neighbor-constrained 4DGS.

| Methods | SSIM ↑ | LPIPS ↓ | FVD-F ↓ | FV4D ↓ |
|---|---|---|---|---|
| w/o Binary Gate | 0.987 | 0.015 | 78.218 | 57.284 |
| w/o Neighbor Loss | 0.986 | 0.017 | 83.421 | 61.324 |
| Full setting | **0.987** | **0.015** | **73.284** | **55.472** |

### B.2  CHARACTERSHOT VS. TWO-STAGE 4D GENERATION

Experiments in Section 4.2 have demonstrated that CharacterShot significantly outperforms other single-view video-driven 4D generation methods (Xie et al., 2025; Yang et al., 2025b; Zeng et al., 2025). To comprehensively explore the advantages of CharacterShot over existing 4D methods in two-stage generation, we extend the single-view videos from the original MimicMotion and the ground truth for comparison and conduct the ablation study on L4GM.

Figure 9: Visual comparison of 3D multi-view image synthesis. Finetuning SV3D on the Character4D dataset, our view generator generates novel character views that are vivid and more detail-oriented.

As shown in Table 7, L4GM achieves better evaluation scores when given ground-truth single-view video as input. However, producing such high-quality and coherent single-view videos through 3D modeling or manual creation is time-consuming and labor-intensive. In contrast, CharacterShot achieves significantly superior performance using only a single reference character and a pose sequence, demonstrating its flexible and effective 4D character animation capability. We also observe that the finetuned MimicMotion outperforms the original model, although it still falls short of the ground-truth videos, demonstrating the fairness of our comparison using the finetuned MimicMotion.

Table 7: Experiments on different types of single-view video inputs for L4GM. "Original" and "Finetuned" refer to single-view video inputs generated using the original or finetuned MimicMotion models, respectively, while "Ground-Truth" refers to the input ground-truth single-view video.

| Methods | SSIM ↑ | LPIPS ↓ | FVD-F ↓ | FV4D ↓ |
|---|---|---|---|---|
| Original | 0.904 | 0.099 | 1198.655 | 1258.118 |
| Finetuned | 0.907 | 0.091 | 1056.498 | 1042.443 |
| Ground-Truth | 0.916 | 0.081 | 901.819 | 922.767 |
| CharacterShot | **0.971** | **0.025** | **368.235** | **406.624** |

## B.3 CHARACTER DATASETS

We evaluate the effectiveness of our proposed Character4D by comparing our finetuned view generator with the base model SV3D (Voleti et al., 2025) and other SOTA methods such as Zero123XL (Liu et al., 2023), InstantMesh (Xu et al., 2024), and Hi3D (Yang et al., 2024a). Visualizations in Figure 9 demonstrate that

Table 8: Experiments of view images generation on CharacterBench between SOTA methods and our finetuned view generator.

| Methods | PSNR ↑ | SSIM ↑ | LPIPS ↓ | FID ↓ | CLIP-S ↑ |
|---|---|---|---|---|---|
| Hi3D | 18.279 | 0.922 | 0.073 | 77.351 | 94.184 |
| Zero123xl | 15.704 | 0.889 | 0.112 | 78.855 | 93.149 |
| InstantMesh | 17.011 | 0.878 | 0.087 | 76.623 | 92.824 |
| SV3D | 17.340 | 0.906 | 0.153 | 103.543 | 88.499 |
| CharacterShot | **21.098** | **0.945** | **0.054** | **71.656** | **94.513** |

our view generator achieves superior performance in preserving character details for different views—such as facial features, hair, and body structure—compared to other baselines. Experiments in Table 8 also highlights the necessity of the character-centric dataset for multi-view images generation.

## B.4 USER STUDY ON OUT-OF-CHARACTER4D TEST SAMPLES

To evaluate the CharacterShot's generalize ability to characters that are out-of-Character4D (OOC), we construct a test set, which includes characters sourced from the Internet and Flux, spanning 2D anime characters, real-world humans, and other distinct 3D models with diverse motions, to compare CharacterShot with the 4D baselines. Since ground-truth

Table 9: User Study on characters that are OOC.

| Methods | Appearance ↑ | Pose ↑ | Time ↑ | View ↑ |
|---|---|---|---|---|
| SC4D | 21.79 | 19.99 | 21.04 | 20.77 |
| STAG4D | 18.80 | 16.50 | 17.10 | 19.59 |
| L4GM | 12.22 | 17.76 | 17.41 | 12.29 |
| DG4D | 7.91 | 16.77 | 14.15 | 10.30 |
| CharacterShot | **39.24** | **29.01** | **30.33** | **37.05** |

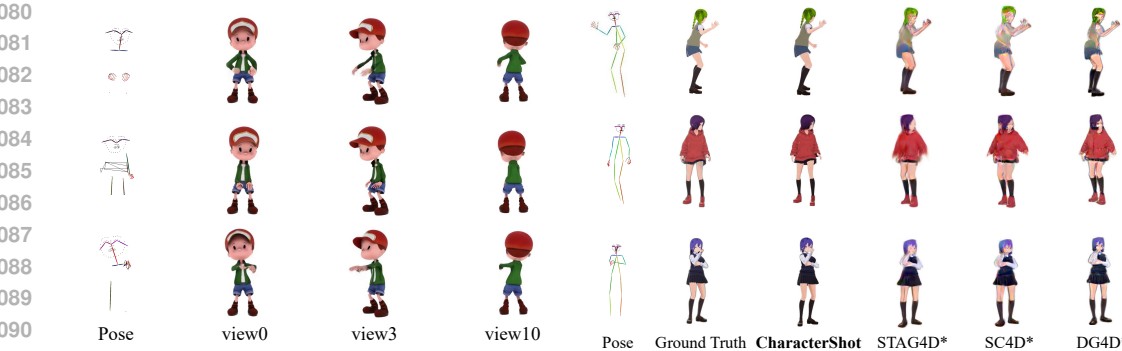

Figure 10: Visualization when Character-Shot meets inaccurate poses.

Figure 11: Visual comparison with finetuned baselines (represents as *).

multi-view videos aren't available for these OOC characters, we conduct a user study with 30 volunteers to assess consistency in appearance, pose, time, and view in Table 9. CharacterShot generalize well to these OOC characters and motions, outperforming all baselines on the OOC test set. Multi-view videos and 4d demos are shown in Supplementary Material.

### B.5 INFERENCE COST

CharacterShot requires 20 or 40 minutes and 37 GB or 8 GB of VRAM to generate multi-view videos on a single H800 GPU, depending on whether CPU-offload is used. The 4DGS stage takes 30 minutes for optimization. While a standard CGI pipeline—including 3D modeling, motion capture, rigging, and more—typically takes several weeks, CharacterShot offers a low-cost CGI solution for individual creators on consumer-grade GPUs.

## C LIMITATION

Although CharacterShot improves robustness to varied pose sequences through confidence-aware pose guidance, which uses the brightness of keypoints and limbs to encode pose-estimation confidence. As shown in Figure 10, CharacterShot performs well and produces robust, stable results in cases where poses disappear (row 1), are disrupted (row 2), or overlap (row 3), thanks to its confidence-aware pose-guidance strategy. However, animating with significantly inaccurate poses remains challenging, highlighting direction for future exploration.

Table 10: Quantitative results for finetuned Zero123xl and Stable-Zero123 on Character-Bench (* represents finetuned models).

| Methods | PSNR ↑ | SSIM ↑ | LPIPS ↓ | FID ↓ |
|---|---|---|---|---|
| Zero123xl | 15.704 | 0.889 | 0.112 | 78.855 |
| Stable-Zero123 | 16.462 | 0.893 | 0.010 | 104.043 |
| Zero123xl* | **22.049** | **0.931** | **0.005** | **50.076** |
| Stable-Zero123* | **20.632** | **0.942** | **0.004** | **45.268** |

## D FINETUNE BASELINES ON CHARACTER4D

In this section, we finetune the baseline methods, STAG4D, SC4D, and DG4D, on our Character4D dataset by training the prior diffusion models Zero123-XL and stable-Zero123 (Liu et al., 2023) used in these baselines. First, the results in Table 10 show that the finetuned Zero123-XL achieve superior performance on characters compared to their raw versions. Next, we evaluate the baseline methods built on these finetuned prior diffusion models and report the qualitative and quantitative results on the CharacterBench in Figure 11 and Table 11.

Table 11: Quantitative comparison between finetuned baseline methods and CharacterShot on CharacterBench (* represents finetuned models).

| Methods | SSIM ↑ | LPIPS ↓ | FVD-F ↓ | FV4D ↓ |
|---|---|---|---|---|
| SC4D* | 0.914 | 0.090 | 1072.756 | 1093.035 |
| STAG4D* | 0.919 | 0.084 | 1028.930 | 985.809 |
| DG4D* | 0.918 | 0.081 | 950.306 | 866.897 |
| CharacterShot | **0.971** | **0.025** | **368.235** | **406.624** |

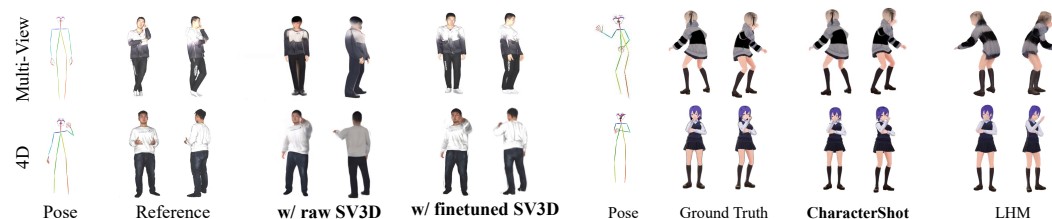

Figure 12: Visualization for CharacterShot with raw SV3D or finetuned SV3D.

Figure 13: Visual comparison with animatable 3DGS method LHM.

CharacterShot outperforms all baselines when they are also finetuned on our Character4D dataset. We also conduct evaluations on the real-human datasets People-Snapshot and THuman 2.1 using CharacterShot with and without finetuned SV3D in Figure 12, CharacterShot with the raw SV3D produces results with blurred details in both the facial and body regions, whereas finetuning SV3D yields more consistent and higher-quality results.

# E  COMPARISON WITH ANIMATABLE 3DGS

In this section, we compare CharacterShot with rencent animatable 3DGS method LHM (Qiu et al., 2025a) on our CharacterBench. As shown in Figure 13, CharacterShot achieves more precise pose alignment and higher-quality facial and body details across different views. The

Table 12: Quantitative comparison between animatable 3DGS method LHM and CharacterShot on CharacterBench.

| Methods | SSIM ↑ | LPIPS ↓ | FVD-F ↓ | FV4D ↓ |
|---|---|---|---|---|
| LHM | 0.933 | 0.072 | 883.416 | 847.143 |
| CharacterShot | **0.971** | **0.025** | **368.235** | **406.624** |

quantitative results in Table 12 further demonstrate that CharacterShot outperforms LHM on all metrics. Animatable 3DGS methods such as LHM require less computation time but sacrifice motion accuracy and fine-grained reconstruction of human details. In contrast, our 4DGS pipeline leverages the powerful generative capability of diffusion models to achieve more precise and flexible motion control, and optimizes more consistent, higher-quality 4DGS representations from multi-view videos that provide rich geometric, appearance, and motion information, while incurring higher computational cost than animatable 3DGS methods. We would like to clarify that 4DGS and animatable 3D are two distinct approaches to 4D animation. Moreover, 4DGS remains valuable for applications such as the metaverse, camera production, and city reconstruction, and we should not discontinue exploring it solely because of its current drawbacks (e.g., it is more time-consuming than animatable 3DGS), as these limitations are likely to be addressed by future advances in algorithms and hardware.

# F  THE USE OF LARGE LANGUAGE MODELS (LLMS)

We use large language models (LLMs) solely for the limited purpose of checking grammar and polishing the overall writing style of our texts. Their role is restricted to improving readability, fluency, and correctness, rather than contributing to the generation of new ideas or altering the substance of our work. By focusing only on surface-level language refinement, we ensure that the originality, logical structure, and core arguments of the content remain entirely authored by us. In this way, LLMs serve as supportive tools for linguistic clarity, not as creators or co-authors of intellectual contributions.

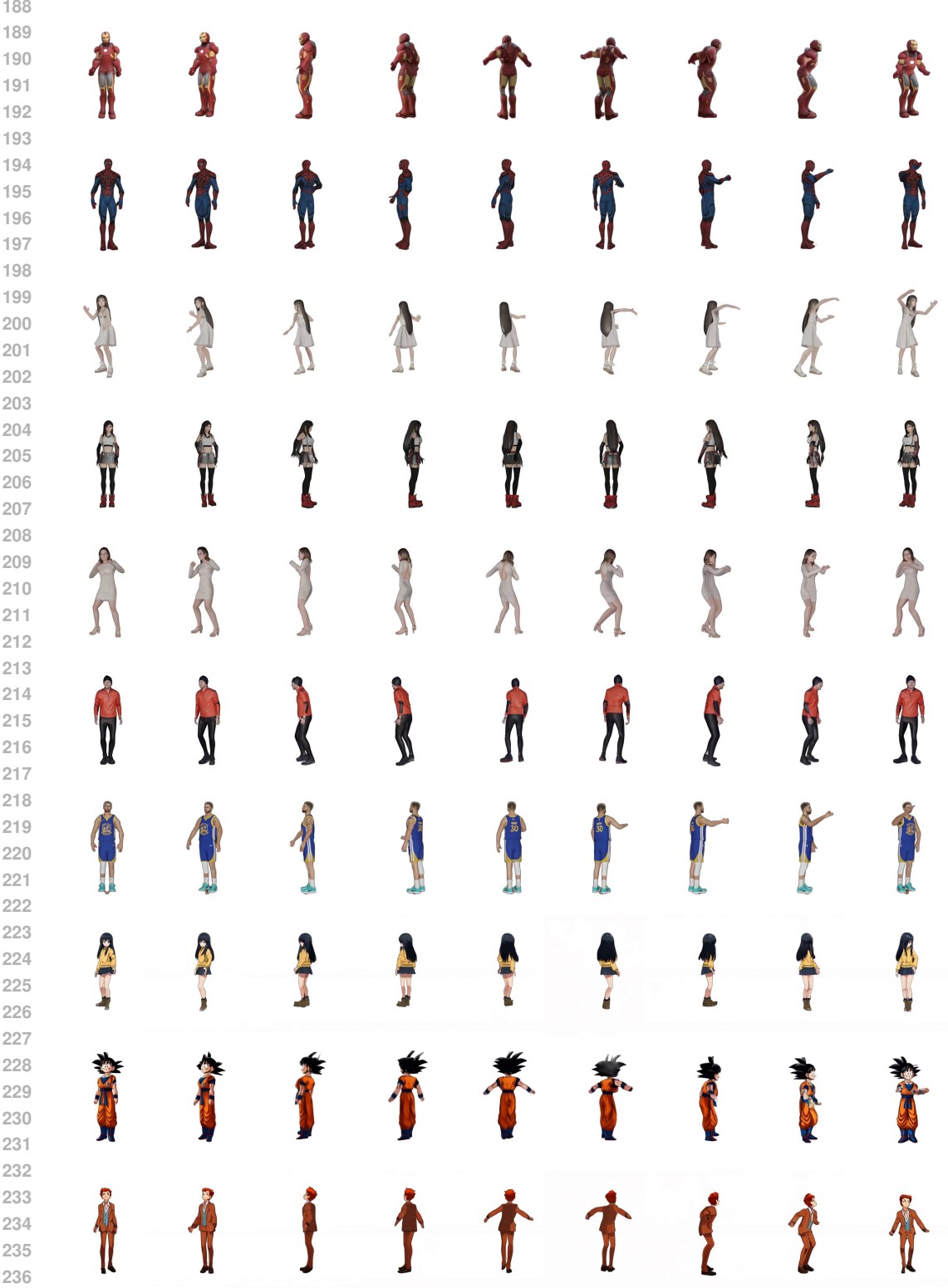

Figure 14: Visual results of multi-view videos generation for characters from Flux and Internet, which are out-of-Character4D. Specifically, Iron Man, Spider-Man, and Tifa (from Final Fantasy) in rows 1–4 are characters from modern games or movies. Rows 5–7 show real-world humans, and rows 8–10 show 2D anime characters.

