# OpenReview forum: "CharacterShot: Controllable and Consistent 4D Character Animation"
_ICLR.cc/2026/Conference — Submitted to ICLR 2026_

### Official Review · Reviewer_RGZ3 · 2025-10-16

**Soundness:** 2
**Presentation:** 2
**Contribution:** 3
**Rating:** 4
**Confidence:** 3

**Summary:**

This paper introduces CharacterShot, a controllable 4D character animation framework that generates multi-view consistent 3D characters from a single reference image and a 2D pose sequence. Built upon the DiT-based image-to-video model CogVideoX, CharacterShot enables user-defined motion control and extends 2D animation to 3D by incorporating a dual-attention module and camera priors to ensure spatio-temporal and cross-view consistency. To create stable 4D outputs, the method applies a neighbor-constrained 4DGS optimization. The authors also present Character4D, a large-scale dataset and promised to open-source.

**Strengths:**

CharacterShot introduces a novel framework that generates 4DGS from a single reference image and a 2D pose sequence. It combines a dual-attention module and camera priors to ensure both spatial-temporal and cross-view consistency. The proposed neighbor-constrained 4D Gaussian Splatting further improves geometric stability and reduces artifacts in motion.

Additionally, the introduction of the large-scale Character4D dataset significantly advances research resources for future 4D character animation. The effectiveness of constructed dataset is shown with the comparison with SV3D.

**Weaknesses:**

CharacterShot adopted 4DGS rather than animatable 3DGS, which introduces practicality concerns.

Generating a separate 4DGS for every motion sequence is computationally expensive and time-consuming, which CharacterShot require 20+30 minutes in H800 GPU. In contrast, animatable 3DGS methods such as LHM[1] and AniGS[2] can model dynamic motion through deformation fields or skeletal animation, enabling efficient reuse of a single canonical representation across sequences.

Furthermore, the paper lacks direct comparison with recent animatable 3DGS methods such as LHM and AniGS, which are more suitable baselines for evaluating motion-controllable representations. This omission limits the fairness and completeness of the evaluation, leaving open whether CharacterShot’s 4D generation truly offers superior efficiency or flexibility in real-world animation scenarios.

In summary, my major concern is (1) What is the benefit of generating in 4DGS rather than animatable 3DGS (2) and the lack of comparison with those methods.

[minor]
There are few errors in citation.

L69 :  Objverse Deitke et al. (2023) > Objverse (Deitke et al., 2023)

L544 : missing year for Cameractrl


[1] LHM: Large animatable human reconstruction model from a single image in seconds. arXiv preprint arXiv:2503.10625.

[2] AniGS: Animatable gaussian avatar from a single image with inconsistent gaussian reconstruction. CVPR 2025

**Questions:**

L990–994: Is the H800 a consumer-grade GPU?

How long can CharacterShot generate? Is it fixed?

---

> ### Author Response · Authors · 2025-11-23
> **Rebuttals from authors**
>
> > W1: Comparison with animatable 3DGS.
>
> Since AniGS is not open source, we compare CharacterShot with rencent animatable 3DGS method LHM[1] on our CharacterBench.
> As shown in Figure 13, CharacterShot achieves more precise pose alignment and higher-quality facial and body details across different views.
> The quantitative results in Table 12 further demonstrate that CharacterShot outperforms LHM on all metrics.
>
> 4DGS and animatable 3DGS are two distinct approaches to 4D animation.
> Current 4DGS pipelines leverage diffusion priors to achieve more precise and flexible motion control, and optimizes more consistent, fine-grained 4DGS representations, which remain valuable for applications such as the metaverse and city reconstruction.
> We **should not discontinue exploring it solely because of its current drawbacks** (e.g., it is more time-consuming than animatable 3DGS), as these limitations are likely to be addressed by future advances in algorithms and hardware.
>
> [1]LHM: Large animatable human reconstruction model from a single image in seconds, arXiv 2025.
>
> > W2: Typos.
>
> Thank you for pointing out these typos. We have corrected them in the revised manuscript.
>
> > Q1: Is the H800 a consumer-grade GPU?
>
> No. But CharacterShot only requires 8 GB of VRAM with cpu offload, which is enough to run on a consumer-grade GPU.
>
> > Q2: Longer generation.
>
> CharacterShot generates 8N+1 frames (where N<=6), following its base I2V model CogVideoX-5B-I2V.
> In the revised Supplementary Material, we provide multi-view videos with 49 frames.

---

> ### Author Response · Authors · 2025-11-27
> **Seeking Further Feedback**
>
> Dear Reviewer RGZ3:
>
> Again, thank you very much for the detailed comments.
>
> **We hope that our rebuttal could address your questions and concerns**, such as 'Comparison with animatable 3DGS' and 'Longer generation'.  As the author-reviewer discussion phase is concluding, **we kindly ask you to review our revised paper and our response and consider adjusting the scores if our response has addressed all your concerns.** Otherwise, please let us know if there are any other questions. **We would be more than happy to answer any further questions.**
>
> Best regards,
>
> The Authors

---

### Official Review · Reviewer_7CK3 · 2025-10-25

**Soundness:** 2
**Presentation:** 2
**Contribution:** 2
**Rating:** 4
**Confidence:** 4

**Summary:**

This paper introduces CharacterShot, a novel 4D character animation framework aiming to "democratize" the CGI pipeline. It enables individual creators to generate controllable and consistent 4D (i.e., dynamic 3D) character animations from a single character reference image and a 2D pose sequence. The framework is a multi-stage pipeline: First, it pretrains a DiT-based 2D video model to accurately follow input 2D poses. Second, it lifts this 2D model to 3D by introducing a novel "dual-attention module" and camera priors to generate multi-view videos with spatio-temporal and cross-view consistency. Finally, it employs a "neighbor-constrained 4D Gaussian Splatting (4DGS)" optimization method to reconstruct a continuous and stable 4D character representation from these generated videos. To support this task, the authors also contribute a large-scale dataset, Character4D (with 13,115 characters), and a new benchmark, CharacterBench. Experiments demonstrate that the method outperforms existing SOTA approaches on the authors' self-constructed benchmark.

**Strengths:**

Novel and Practical Problem Formulation: Generating 4D animation from a single image and 2D poses is a highly challenging yet valuable task. It significantly lowers the barrier to 4D content creation, as its input requirements are far less restrictive than methods requiring multi-view videos, 3D models, or even single-view videos. This has strong practical application potential.

Solid Technical Contributions: The paper proposes a complete and technically sound pipeline to address this complex problem. The "dual-attention module" for simultaneously modeling spatial-temporal and spatial-view consistency is a key innovation for lifting 2D to multi-view 3D. Furthermore, the "neighbor-constrained 4DGS" directly tackles the stability issues inherent in 3D reconstruction from AI-generated videos, which may suffer from noise or inconsistencies.

**Weaknesses:**

Self-Serving Evaluation on a Niche Benchmark: A major weakness is that all quantitative comparisons rely exclusively on the authors' newly created CharacterBench, which is built from their own Character4D dataset. While dataset contribution is noted, this creates a circular evaluation loop where the method is tested on the same data distribution it was trained on (or at least a very similar one, derived from VRoidHub). This benchmark, filled with 13k anime-style characters, may not be representative of broader 4D animation challenges (e.g., real humans, complex physics) and makes it impossible to fairly compare against SOTA methods, which were tuned for different data and tasks.

Questionable Generalization and Robustness: While the appendix shows generalization to "Out-of-Character4D" samples (real humans, other 3D models), the main evaluation is heavily focused on anime-style characters from VRoidHub. The method's performance on real-world humans, complex clothing dynamics (like capes or skirts), and its robustness to inaccurate pose estimations (which the authors admit as a limitation) are not thoroughly or quantitatively validated.

System Complexity and Potential Error Propagation: The entire framework is a cascaded system (2D Animation -> Multi-view Video Gen -> 4DGS Optimization). This multi-stage pipeline is highly complex and relies on multiple fine-tuned models (e.g., CogVideoX and SV3D). This implies that errors can propagate and accumulate: for instance, if the multi-view video generation in Stage 2 is of low quality or lacks consistency, the 4DGS optimization in Stage 3 will likely fail to recover a high-quality 4D result, even with the neighbor-constraint.

**Questions:**

Regarding Generalization: The Character4D dataset consists mainly of anime-style characters. How would CharacterShot perform if trained or tested on broader datasets focused on real humans (e.g., People-Snapshot, THHuman 2.0, or other dynamic human capture datasets)? This is crucial for understanding the model's ability to generalize to realistic human morphology and complex textures.

Regarding the Dual-Attention Module: Could the authors elaborate on the design of the "dual-attention module"? How does it fundamentally differ, architecturally and computationally, from simply using separate (or sequential) spatial, temporal, and view-attention mechanisms (as critiqued in SV4D)? Why is this design more effective at capturing "implicit visual transmission"?

Regarding Dependency on Intermediate Stages: The framework appears to rely on a fine-tuned SV3D as a "View Generator" to provide reference images for the multi-view video generation. To what extent is the final 4D animation quality sensitive to the fidelity of this initial static multi-view generation? If the view generator fails (e.g., produces an incorrect view or distorted details), can the system handle this robustly?

---

> ### Author Response · Authors · 2025-11-23
> **Rebuttals from authors**
>
> > W1: Finetuned baselines on Character4D:
>
> Please see General Response #1.
>
> > W2 & Q1: Evaluation on real humans.
>
> Please see General Response #2.
>
> > W3: Robustness to inaccurate pose:
>
> We show the visual results when CharacterShot encounters inaccurate poses in Figure 10.
> CharacterShot performs well and produces robust, stable results in cases where poses disappear (row 1), are disrupted (row 2), or overlap (row 3), thanks to its confidence-aware pose-guidance strategy.
> However, improving robustness to such severely inaccurate poses is a common challenge for all pose-driven methods and is not the main focus of this paper.
> We leave this for future exploration.
>
> > W4: Accumulation error:
>
> 2D Animation is a pretraining stage and is not included during inference.
> This two-stage pipeline (multi-view video generation → 4DGS optimization) has become a standard paradigm for more consistent and higher-quality 4D generation, as demonstrated in recent works such as SV4D[1] and Diffusion2[2].
> Each stage is a plug-and-play module, which makes it easier to localize errors and allows components to be replaced with more powerful models without additional cost.
> CharacterShot follows this paradigm and introduces a dual-attention mechanism and a neighbor-constraint optimization to reduce potential error accumulation.
>
> [1]SV4D: Dynamic 3D Content Generation with Multi-Frame and Multi-View Consistency, ICLR 2025.
>
> [2]Diffusion2: Dynamic 3D Content Generation via Score Composition of Video and Multi-view Diffusion Models, ICLR 2025.
>
> > Q2: Details of dual-attention module:
>
> The key architectural difference is that the separate attentions are implemented as 2D attention, where each attention is restricted to interactions within a single like spatial, temporal, and view.
> As shown in Figure 3, the blue patches (view attention) and yellow patches (temporal attention) fail to model the left and right hands across different views and frames, which represents the **implicit visual transmission**.
> In contrast, our dual-attention module employs a 3D full-attention mechanism, which jointly models interactions across two dimensions.
> When we apply 3D full attention to the spatial–temporal and spatial–view branches, each patch can attend to all patches across views and frames, thereby capturing the **implicit visual transmission**.
> Also, the two parallel 3D full-attention blocks in our dual-attention module have fewer trainable parameters (about two-thirds as many) but incur higher computational cost than three separate 2D attention blocks.
>
> > Q3: Concerns about view-generator.
>
> The performance of CharacterShot is influenced by the quality of the input multi-view images, so we fine-tune SV3D to achieve more stable and robust multi-view image generation.
> As shown in Figure 12, CharacterShot with raw SV3D achieves inferior performance (including blurred regions in the face and body) compared to using a fine-tuned SV3D on real-human images.
> Moreover, we treat view-generator as the a plugin component in our work followed SV4D that allows us to seamlessly replace SV3D with any more powerful view-generator at no additional cost.

---

> ### Author Response · Authors · 2025-11-27
> **Seeking Further Feedback**
>
> Dear Reviewer 7CK3:
>
> Again, thank you very much for the detailed comments.
>
> **We hope that our rebuttal could address your questions and concerns**, such as ' Comparison with baselines finetuned on Character4D', 'Evaluation on real humans' and 'Accumulation error'.  As the author-reviewer discussion phase is concluding, **we kindly ask you to review our revised paper and our response and consider adjusting the scores if our response has addressed all your concerns.** Otherwise, please let us know if there are any other questions. **We would be more than happy to answer any further questions.**
>
> Best regards,
>
> The Authors

---

### Official Review · Reviewer_MbSK · 2025-10-26

**Soundness:** 2
**Presentation:** 3
**Contribution:** 2
**Rating:** 6
**Confidence:** 3

**Summary:**

The paper presented a method, referred as CharacterShot, to create dynamic 3D characters from a single reference character image and a 2D pose sequence. Given a reference character image, CharacterShot employs the I2V model CogVideoX to generate its corresponding video sequence controlled by pose conditions. The paper proposed a dual-attention module to improve the spatio-temporal and cross-view consistency in the video. Afterwards, CharacterShot employs a coarse-to-fine 4D Gaussian Splatting to fix artifacts in the multi-view videos. The paper also presented a new Character4D animation dataset including 13,115 unique characters to fine-tune the SV3D model to generate multi-view images in the CharacterShot method. A separate CharacterBench is used to evaluate and compare with 4 other single-view video-driven 4D generation methods.

**Strengths:**

The visual quality of the multi-view character videos generated by CharacterShot appears clean and consistent, and very close to the ground truth.

The dual-attention module, which uses parallel 3D full attention blocks to enforce visual consistency across spatial-temporal multi-view images, is an interesting and novel approach.

The coarse-to-fine 3DGS, including the neighbor constraints in the fine stage, appears a reasonable post-processing to improve the character video.

**Weaknesses:**

Since CharacterShot employs the I2V model CogVideoX that is DiT-based to generate the video given a character image, the paper claims this is the first DiT-based 4D character animation work. This claim is not a well-supported one to me.

The major experiments are only conducted on the new CharacterBench dataset introduced in this paper. The fairness of the comparison with 4 other methods on the this CharacterBench needs further justification, e.g., if other methods have been fine-tuned on the Character4D dataset as well.

**Questions:**

What are the essential difference between generating multi-view videos from a character image or a human figure image? Which task is more challenging?

The proposed CharacterShot framework does not seem to be restricted to \emph{character} video generation, except that the multi-view images are generated by the SV3D model fine-tuned on the Character4D dataset. So, if using a SV3D not finetuned on Character4D and employing a reference image of a real person, what the performance of the proposed method would be and compared to many latest relevant works? The technical advantages of the proposed method, rather than using dedicated character datasets, would be validated on other human figure video generation tasks.

The sample character images shown in the paper do not show that much diversity. Any more sophisticated characters in high-resolution, e.g., NPC in modern games, are tested?

A minor suggestion: if some references have been published in previous ICLR, I would suggest to cite their ICLR version instead of the arXiv version.

---

> ### Author Response · Authors · 2025-11-23
> **Rebuttals from authors**
>
> > W1: The claim is not well supported.
>
> We would like to clarify that our claim is that CharacterShot is the first work to enable **2D pose-driven multi-view video generation** based on a DiT model, with an additional robust **4D optimization stage**.
> Previous methods like Diffusion2[1], SV4D[2] generate multi-view videos from a single-view video using U-Net-based models.
> Human4DiT[3] employs complex and computationally expensive SMPL models as motion control within a DiT framework and omits the 4D optimization stage.
> Moreover, these methods target real-world scenarios and do not generalize well to characters.
>
> [1]Diffusion2: Dynamic 3D Content Generation via Score Composition of Video and Multi-view Diffusion Models, ICLR 2025.
>
> [2]SV4D: Dynamic 3D Content Generation with Multi-Frame and Multi-View Consistency, ICLR 2025.
>
> [3]Human4DiT: 360-degree Human Video Generation with 4D Diffusion Transformer, SIGGRAPH ASIA 2024.
>
> > W2: Finetune baselines on Character4D.
>
> Please see General Response #1.
>
> > Q1: Difference between character image and human image.
>
> Humans share similar body proportions and appearance space, allowing models to easily exploit this structural prior and perform well on humans.
> In contrast, characters are editable and flexible, embodying the creators’ imagination and spanning highly diverse and distinct proportions and appearances like exaggerated big head or eyes and Iron Man's armor.
> Generating multi-view videos from a character image requires the model to capture the inherent relationship between the spatial-temporal and spatial-view, making the task of characters more challenging.
>
> > Q2: Inference on raw SV3D and evaluation on real persons.
>
> We conduct evaluations on the real-human datasets People-Snapshot[1] and THuman 2.1[2] using CharacterShot (with raw SV3D or finetuned SV3D) and baseline methods.
> As shown in Figure 12, CharacterShot with the raw SV3D produces results with blurred details in both the facial and body regions, whereas finetuning SV3D yields more consistent and higher-quality results.
> We also present the comparisons on real persons between CharacterShot (with finetuned SV3D) and the baseline methods in Figure 5, Figure 6 and Table 3, which demonstrate CharacterShot’s strong generalization ability to real persons and its superior performance over the baselines.
>
> [1]Video Based Reconstruction of 3D People Models, CVPR 2018.
>
> [2]Function4D: Real-time Human Volumetric Capture from Very Sparse Consumer RGBD Sensors, CVPR 2021.
>
> > Q3: Diverse character images.
>
> We have tested a diverse set of characters and present the results in the revised Figure 14.
> Specifically, Iron Man, Spider-Man, and Tifa (from Final Fantasy) in rows 1–4 are characters from **modern games or movies**.
> Rows 5–7 show **real-world humans**, and rows 8–10 show **2D anime characters**.
>
> > Q4: Suggestion for the citation format.
>
> Thank you for pointing out this formatting issue. We have checked and corrected all arXiv citations to their corresponding conference or journal versions as thoroughly as possible.

---

> ### Author Response · Authors · 2025-11-27
> **Seeking Further Feedback**
>
> Dear Reviewer MbSK:
>
> Again, thank you very much for the detailed comments.
>
> **We hope that our rebuttal could address your questions and concerns**, such as 'Claim is not well supported', ' Comparison with baselines finetuned on Character4D', 'Difference between characters and humans' and 'Evaluation on real persons'.  As the author-reviewer discussion phase is concluding, **we kindly ask you to review our revised paper and our response and consider adjusting the scores if our response has addressed all your concerns**. Otherwise, please let us know if there are any other questions. **We would be more than happy to answer any further questions.**
>
> Best regards,
>
> The Authors

---

### Official Review · Reviewer_peLs · 2025-10-31

**Soundness:** 2
**Presentation:** 3
**Contribution:** 3
**Rating:** 6
**Confidence:** 3

**Summary:**

This paper proposes CharacterShot, a novel framework that takes a 2D character image and a 2D pose sequence to generate a 4D animated character. This leverages the I2V model fine-tuned to take 2D pose sequences and camera parameters and output multi-view videos of the character, and then trains a 4DGS model to represent character animation in 4D. A dual 3D full-attention mechanism that applies to the view-spatial dimension and temporal-spatial dimension is used to ensure multi-view consistent video generation. Neighbor-based regularization in the multi-view video for 4DGS optimization enforces geometric consistency. Trained with the Character4D dataset, the method demonstrates superior generation results compared to prior methods using I2V in a single view and creating 4D from it.

**Strengths:**

* A novel framework that takes a 2D character image and a 2D pose sequence for 4D generation sounds promising, as the 2D input is more convenient than typical input while providing decent control over the generated motion.
* The result demonstrates superior multi-view consistency for the generated 4D character animation.
* A novel dataset built for the novel framework allows for further exploration of the idea.

**Weaknesses:**

* Single-view video to 4D baselines, not just 2D to video part, could also be fine-tuned for fair comparison. Their quality degradation is more noticeable in novel views, which may be due to a lack of training data for generating multi-view character videos from single-view character videos.
* Related works discuss prior works with too much focus on the general trend of generation methods. A better summary and greater emphasis on highly relevant works scattered across different sections on character-focused 3D/4D generation (with motion control) would be beneficial.

**Questions:**

* Please clarify the test split used for CharacterBench and the train split used by the Character4D dataset: are motions or characters shared across splits, or are they from a similar source? This can impact the validity of the benchmark result.

* L234: for a each -> for each
* L297: The group seems to be neighbors, and how those neighbors are selected for each splat could be clarified here.

---

> ### Author Response · Authors · 2025-11-23
> **Rebuttals from authors**
>
> > W1: Finetune baselines on Character4D.
>
> Please see General Response #1.
>
> > W2: Character-Focused 3D/4D Generation.
>
> We discuss the related works about character-focused 3D/4D generation in the Section 2.4 of the revised paper, as well as in the following:
>
> *Focusing on character-centric 3D/4D generation, many methods learn canonical 3D Gaussian (or mesh) representations with pose-driven deformations, either by optimizing them directly from monocular videos (gaussianbody,3dgsavatar,hugs,gart,gaussianavatar) or by predicting them in a feed-forward manner from one or a few images (lhm,pflhm,idol), in order to construct animatable human avatars by binding them to SMPL models.
> With the rapid development of large diffusion models, some works (charactergen,adahuman,anigs,persona,disco4d,ponimator) leverage multi-view or video diffusion priors to generate pose- and view-rich supervision for human and character avatars, enabling the optimization of 3D/4D Gaussian representations with motions from rigged skeletons or bound SMPL models.
> However, these methods generate dynamic 3D characters by deforming static canonical avatars along pre-defined motion trajectories within a rigging and rendering pipeline that is complex, tightly coupled, and difficult for individual users.
> To provide a more user-friendly solution, we propose CharacterShot, which generates high-quality 4D character animation from only a single reference character image and a 2D pose sequence.*
>
> > Q1: Split of training and test set.
>
> We use 13,092 of the 13,115 characters from Character4D as our training dataset, and the remaining 23 characters, together with an additional 10 Mixamo characters, as our test dataset.
> The test characters from Character4D share similar motion and character distributions with the training dataset, whereas the test characters from Mixamo have different motion and character distributions with the training dataset.
>
> For a more valid evaluation, we conduct the evaluation on test characters that are out of Character4D's (OOC) distribution, collecting from InterNet, and generated a suite of virtual characters using Flux, spanning 2D anime characters, other distinct 3D models, and real-world human datasets People-Snapshot[1] and THuman 2.1[2] (following the suggestion of Reviewer 7CK3) with diverse motions.
> Experiments in Figure 5, Figure 6 and Table 3 (real-world human datasets), as well as Figure 14 and Table 9 (other OOC characters), demonstrate that CharacterShot generalizes well to these OOC characters and motions and outperforms all baselines.
>
> [1]Video Based Reconstruction of 3D People Models, CVPR 2018.
>
> [2]Function4D: Real-time Human Volumetric Capture from Very Sparse Consumer RGBD Sensors, CVPR 2021.
>
> > Q2: Typo.
>
> Thanks for your reminder. We have revised this typo in the new version.
>
> > Q3: Details about neighbors selection.
>
> We select 20 nearest neighbors for each point from the static 3D Gaussians based on the L2 distance.

---

> ### Author Response · Authors · 2025-11-27
> **Seeking Further Feedback**
>
> Dear Reviewer peLs:
>
> Again, thank you very much for the detailed comments.
>
> **We hope that our rebuttal could address your questions and concerns**, such as ' Comparison with baselines finetuned on Character4D', 'Related work about Character-Focused 3D/4D Generation' and 'Training and test split'.  As the author-reviewer discussion phase is concluding, **we kindly ask you to review our revised paper and our response and consider adjusting the scores if our response has addressed all your concerns**. Otherwise, please let us know if there are any other questions. **We would be more than happy to answer any further questions.**
>
> Best regards,
>
> The Authors

---

### Author Response · Authors · 2025-11-23
**Overall Comments**

Thank all reviewers for reviewing and providing constructive feedbacks to our paper. We also deeply appreciate the reviewers’ acknowledgment of:

- **Novelty** of proposed CharacterShot, including dual-attention and neighbor-constrained 4DGS optimization. (reviewers peLs, MbSK, RGZ3 and 7CK3)

- Consistent, high-quality and stable results. (reviewers peLs, MbSK and RGZ3)

- Proposed large-scale Character4D dataset. (reviewers peLs and RGZ3)

Please note that we **have revised the manuscript and the supplemental materials, with all changes marked in red**. The main changes are summarized as follows:

- The discussion for the related work about 3D/4D Character Generation in Section 2.4. (reviewer peLs)
- The evluations on real humans in Figure 5, Figure 6, Table 3 and Section 4.2. (reviewers peLs, MbSK and 7CK3)
- The comparisons between CharacterShot and baseline methods finetuned on our Character4D dataset in Figure 11, Table 11 and Appendix D. (reviewers peLs, MbSK and 7CK3)
- The comparison and discussion between animatable 3DGS (LHM[1]) and 4DGS (CharacterShot) in Figure 13, Table 12 and Appendix E. (reviewer RGZ3)

We respond to each reviewer below to address their specific concerns. And for common concerns shared by multiple reviewers, including *Finetune baselines on Character4D* and *Evaluation on real humans*, we provide General Responses in the global comments. Please take a look and let us know if further clarification / discussion is needed. Also, we will include any further discussions in the next version and release codes, models and data of CharacterShot.

[1]LHM: Large animatable human reconstruction model from a single image in seconds, arXiv 2025.

---

> ### Author Response · Authors · 2025-11-23
> **General Response #1 — Finetune baselines on Character4D**
>
> We fine-tune the baseline methods, STAG4D, SC4D, and DG4D, on our Character4D dataset by training the prior diffusion models Zero123-XL[1] and Stable-Zero123[2] used in these baselines.
> First, the results in Table 10 show that the fine-tuned Zero123-XL and Stable-Zero123 achieve superior performance on characters compared to their raw versions.
> Next, we evaluate the baseline methods built on these fine-tuned prior diffusion models and report the qualitative and quantitative results on the CharacterBench in Figure 11 and Table 11.
> CharacterShot outperforms all baselines when they are also fine-tuned on our Character4D dataset.
>
> [1]Zero-1-to-3: Zero-shot One Image to 3D Object, ICCV 2023.
>
> [2]https://huggingface.co/stabilityai/stable-zero123

---

> ### Author Response · Authors · 2025-11-23
> **General Response #2 — Evaluation on real humans**
>
> We evaluate CharacterShot and baseline methods on 48 real humans from People-Snapshot[1] (24 humans) and THuman 2.1[2] (24 humans).
> Please note that there are no ground-truth multi-view videos for these real humans, so we conduct a user study with 30 volunteers to assess consistency in appearance, pose, time, and viewpoint instead of quantitative evaluations.
> Results in Figure 5, Figure 6 and Table 3 demonstrate that CharacterShot generalizes well to these real-world humans and outperforms all baselines.
>
> [1]Video Based Reconstruction of 3D People Models, CVPR 2018.
>
> [2]Function4D: Real-time Human Volumetric Capture from Very Sparse Consumer RGBD Sensors, CVPR 2021.

---

### Meta-Review · Area_Chair_qBUX · 2026-01-05

**Summary:**

This paper proposes CharacterShot, a framework to generate a 4D animated character by taking a 2D character image and a 2D pose sequence. The  fine-tuned I2V model like CogVideoX output multi-view videos of the character, and then trains a 4DGS model to represent character animation in 4D.  Besides, a new dataset Character4D is proposed for training the model.

**Reviewer Concerns:**

1. Misleading and some typos are addressed.
2. Robustness of the proposed method is addressed.
3. Technique contributions are relatively addressed.
4. Self-Serving Evaluation on a Niche Benchmark is not well addressed.

Overall, most of concerns are addressed but the main concern (evaluation) are not well-addressed in rebuttal.

**Reviewer Scores:**

Two reviewers give positive score and the other two reviewers give negative score. I think the two negative comments are hard change based on the rebuttal.

---

### Decision · Program_Chairs · 2026-01-26

Reject